

# Serum lipid level is not associated with symptomatic intracerebral hemorrhage after intravenous thrombolysis for acute ischemic stroke

Ting-Chun Lin[1], Yen-Kuang Lin[2], Chin-I Chen[3], Lung Chan[4], Nai-Fang Chi[4], Rey-Yue Yuan[1], Jau-Jiuan Sheu[1], Chun-Ren Wei[1], Jui-Ping Tsai[1] and Tu-Hsueh Yeh[1]

[1] Department of Neurology, Taipei Medical University Hospital, Taipei Medical University, Taipei, Taiwan
[2] Research Center of Biostatistics, Taipei Medical University, Taipei, Taiwan
[3] Department of Neurology, Wan Fang Hospital, Taipei Medical University, Taipei, Taiwan
[4] Department of Neurology, Shuang Ho Hospital, Taipei Medical University, New Taipei City, Taiwan

## ABSTRACT

**Background.** This study assessed whether serum lipid levels are associated with the risk of symptomatic intracerebral hemorrhage (sICH) and functional outcomes in patients with acute ischemic stroke after receiving intravenous thrombolysis.

**Methods.** We retrospectively analyzed consecutive ischemic stroke patients who were treated with intravenous tissue plasminogen activator between January 2007 and January 2017. Lipid levels on admission, including total cholesterol, low-density lipoprotein cholesterol (LDL-C), high-density lipoprotein cholesterol (HDL-C), and triglyceride levels, as well as potential predictors of sICH were tested using univariate and multivariate analyses.

**Results.** Of the 229 enrolled patients (100 women, aged $68 \pm 13$ years), 14 developed sICH and 103 (45%) had favorable functional outcomes at 3 months. The patients with sICH more often had diabetes mellitus (71% vs. 26%, $P = 0.01$) and had more severe stroke (mean National Institutes of Health Stroke Scale [NIHSS] score of 16 vs. 13, $P = 0.045$). Regarding lipid subtype, total cholesterol, LDL-C, HDL-C, and triglyceride levels were not associated with sICH or functional outcomes. According to the results of multivariate analysis, the frequency of sICH was independently associated with diabetes mellitus (odds ratio [OR] = 6.04; 95% CI [1.31–27.95]; $P = 0.02$) and the NIHSS score (OR = 1.12; 95% CI [1.02–1.22]; $P = 0.01$). A higher NIHSS score was independently associated with unfavorable functional outcomes (OR = 0.86; 95% CI [0.81–0.91]; $P < 0.001$).

**Conclusions.** Serum lipid levels on admission, including total cholesterol, LDL-C, HDL-C, and triglyceride levels, were not associated with sICH or 3-month functional outcomes after intravenous thrombolysis for acute ischemic stroke.

Corresponding author
Ting-Chun Lin, Lteikin@gmail.com, 131004@h.tmu.edu.tw

## INTRODUCTION

Stroke is the second leading cause of death worldwide, and the overall global burden of stroke is high and still increasing (*Feigin et al., 2014*). Several predictors of ischemic stroke have been examined, including age, diabetes mellitus, cardiac disease, hypertension, cardioembolic etiology, and dyslipidemia. The standard of care for acute ischemic stroke comprises endovascular treatment and intravenous thrombolysis (*Bhaskar et al., 2018*). Intravenous thrombolysis with recombinant tissue plasminogen activator (rt-PA) given within 4.5 h of stroke onset is effective in the acute stage of ischemic stroke (*Hacke et al., 2008*). However, the intravenous administration of rt-PA induces a 10-fold increase in symptomatic intracerebral hemorrhage (sICH) of brain infarction through breakdown of the neurovascular unit (*Suzuki, Nagai & Umemura, 2016*). Post-thrombolytic sICH resulted in poor outcomes, with the mortality rate reaching 8% at 3 months (*Strbian et al., 2011*). Thus, identifying the risk of sICH in patients receiving recanalization therapy is crucial. Epidemiology studies have suggested an association of low total cholesterol and low-density lipoprotein cholesterol (LDL-C) levels with the risk of primary intracerebral hemorrhage (ICH) (*Amarenco et al., 2006*; *Sturgeon et al., 2007*). However, the association between serum lipid levels, which comprises total cholesterol, LDL-C, high-density lipoprotein cholesterol (HDL-C), and triglyceride levels, and the occurrence of post-thrombolytic sICH remains controversial (*Bang et al., 2007*; *Makihara et al., 2012*; *Messe et al., 2013*; *Rocco et al., 2012*; *Uyttenboogaart et al., 2008*). Many studies have suggested that Asians have a two times higher risk of ICH than non-Asians. Thus, identifying the risk of sICH following thrombolysis therapy is important in the Asian population (*Shen et al., 2007*; *Van Asch et al., 2010*).

The present study investigated whether serum lipid levels are associated with sICH occurrence and functional outcomes in patients with acute ischemic stroke who received intravenous administration of rt-PA. In addition, we tested the hypothesis that serum lipid levels are associated with sICH after the intravenous administration of rt-PA for acute ischemic stroke.

## METHOD

### Patient selection

We retrospectively analyzed consecutive ischemic stroke patients who received intravenous administration of rt-PA within 24 h of stroke onset at three hospitals affiliated with the Taipei Medical University System in Taiwan between January 2007 and January 2017. This study was approved and the requirement to obtain a signed consent form for all patients was waived by the Institutional Review Board of Taipei Medical University (N201512020). All patients underwent the standard treatment within 3 h of stroke onset. Inclusion and exclusion criteria for intravenous rt-PA treatment were mainly adopted from the National Institute of Neurological Disorders and Stroke protocol and the protocol of the Taiwan Guidelines for the Management of Stroke. Initial stroke severity was assessed using the National Institutes of Health Stroke Scale (NIHSS). The levels of lipid subtypes, namely total cholesterol, HDL-C, LDL-C, and triglycerides, were determined in the fasting state

in morning following admission to prevent the effect of meals on serum lipid levels. Data on demographics, vascular risk factors, blood pressure, time from stroke onset until treatment, hemoglobin A1c level, and serum glucose level on admission (nonfasting, principally obtained during the first assessment in the emergency room) were recorded. We aimed to determine the relationship between serum glucose levels on acute admission and stroke outcomes. The level of hemoglobin A1c and a history of diabetes mellitus were used as surrogates for the baseline serum glucose level.

## Outcome measures

Patients who received treatment were initially administered an intravenous rt-PA dose of 0.6–0.9 mg/kg and then admitted to the neurology intensive care unit for observation according to the protocol of the Taiwan Guidelines for the Management of Stroke. sICH was considered a neurological deterioration, which was defined as an increase in the NIHSS score of >4 points within 36 h following rt-PA treatment, along with the observation of a hematoma on a noncontrast computed tomography scan of the brain (*Hacke et al., 1998*). Functional outcomes at 3 months were assessed using the modified Rankin scale (mRS).

## Statistical analysis

For continuous variables, Fisher's exact test was performed. For continuous variables, independent samples Student's *t* test or the Mann–Whitney U test was used after testing for normality. Previous studies of patients who received intravenous rt-PA for stroke have reported multiple clinical and demographic predictors of sICH (*Lou et al., 2008*). We performed an univariate logistic regression analysis to predict the effect of the following potential predictors on sICH following rt-PA: age, NIHSS score, sex, time until rt-PA treatment, laboratory variables, vascular risk factors, and lipid subtype levels. The independent effect of lipid subtype levels on sICH was investigated using a multivariable logistic regression model, in which the potential predictors of sICH, namely the NIHSS score, serum glucose level, history of diabetes mellitus, and lipid subtype levels, were tested. Given that the total cholesterol level was correlated with lipid subtype levels, we tested each lipid subtype separately in the regression model. Finally, we used a multivariable regression analysis model to examine the relationship between lipid levels and functional outcomes at 3 months. In this model, the following covariates were entered: age, NIHSS score, serum glucose level, history of diabetes mellitus, and lipid subtype levels. Variables were excluded if possible collinearity occurred (defined as Variance Inflation Factor >10). Significance was defined as a *P* value of <0.05. All statistical analyses were performed using SPSS software, version 18.0 for Windows (SPSS, Chicago, IL, USA).

## RESULTS

A total of 229 patients were included in this study. Their baseline demographic characteristics are listed in Table 1. Of the 229 patients, sICH occurred in 14 (6.1%) patients (95% CI [3.0–9.0]). All the variables did not follow a normal distribution, except for the mean systolic blood pressure. The median NIHSS score and a history of diabetes mellitus were significantly associated with the risk of sICH. No difference was observed

**Table 1  Baseline characteristics.**

| | Total<br>N = 229 | sICH<br>N = 14 | no sICH<br>N = 215 | P value |
|---|---|---|---|---|
| Mean age (SD), years | 68 (13) | 72 (10) | 67 (13) | 0.20 |
| Male, n(%) | 129 (56) | 8 (57) | 121 (56) | 0.95 |
| Median NIHSS score [IQR] | 13 [8–18] | 16 [13–22] | 13 [8–18] | 0.045 |
| Median time until treatment [IQR], min | 105 [75–145] | 120 [83–152] | 105 [74–145] | 0.41 |
| Mean systolic blood pressure (SD), mmHg | 168 (27) | 175 (26) | 167 (27) | 0.29 |
| Mean diastolic blood pressure (SD), mmHg | 95 (20) | 88 (19) | 95 (20) | 0.37 |
| Mean admission serum glucose level (SD), mg/dL | 152 (69) | 192 (93) | 149 (66) | 0.08 |
| Mean total cholesterol level (SD), mg/dL | 184 (41) | 176 (40) | 185 (41) | 0.75 |
| Mean HDL-C level (SD), mg/dL | 45 (13) | 44 (18) | 45 (12) | 0.84 |
| Mean LDL-C level (SD), mg/dL | 118 (34) | 102 (29) | 119 (34) | 0.12 |
| Mean triglyceride level (SD), mg/dL | 125 (77) | 135 (85) | 125 (77) | 0.76 |
| Mean prothrombin time-INR level, (SD) | 1.08 (0.72) | 1.02 (0.08) | 1.09 (0.75) | 0.54 |
| Mean creatinine level (SD), mg/dL | 1.07 (0.43) | 1.13 (0.53) | 1.06 (0.42) | 0.74 |
| Mean *hemoglobin* A1c level (SD), % | 6.60 (1.56) | 7.46 (2.25) | 6.46 (1.39) | 0.28 |
| Vascular risk factors | | | | |
|     History of arterial hypertension, n(%) | 178 (78) | 12 (86) | 166 (77) | 0.74 |
|     Diabetes mellitus, n(%) | 66 (29) | 10 (71) | 56 (26) | 0.01 |
|     Hyperlipidemia, n(%) | 84 (37) | 4 (29) | 80 (37) | 0.51 |
|     Atrial fibrillation, n(%) | 95 (42) | 7 (50) | 88 (41) | 0.51 |
|     Coronary heart disease, n(%) | 41 (20) | 3 (21) | 38 (18) | 0.96 |
|     Current smoker, n(%) | 38 (21) | 1 (7) | 37 (17) | 0.46 |
|     Previous stroke, n(%) | 25 (11) | 1 (7) | 24 (11) | 1.00 |
| Mean rt-PA dose | 0.83 (0.13) | 0.86 (0.09) | 0.83 (0.13) | 0.59 |
| Favorable outcome (mRS ≤ 2) | 103 (45) | 0 (0) | 103 (48) | <0.000 |

**Notes.**

HDL-C, high-density lipoprotein cholesterol; IQR, interquartile range; LDL-C, low-density lipoprotein cholesterol; mRS, modified Rankin scale; NIHSS, National Institutes of Health Stroke Scale; rt-PA, recombinant tissue plasminogen activator; sICH, symptomatic intracerebral hemorrhage; SD, standard deviation.

in the occurrence sICH according to LDL, HDL, total cholesterol, and triglyceride levels. The comparison of clinical and laboratory results between patients with and without sICH is shown in Table 2. According to the results of the univariate analysis, the serum glucose level on admission (odds ratio [OR] = 1.01; 95% CI [1.00–1.01]; $P = 0.03$) and a history of diabetes mellitus (OR = 7.00; 95% CI [2.11–23.25]; $P = 0.001$) were significantly associated with the occurrence of sICH (Table 2). The total cholesterol, HDL-C and LDL-C levels appeared to be associated with lower risk of sICH, although not significant (OR = 0.995, 0.997, 0.98, all $P$'s > 0.05). After adjustment for covariates in the multivariable regression model, the NIHSS score (OR = 1.12; 95% CI [1.02–1.22]; $P = 0.01$) and a history of diabetes mellitus (OR = 6.04; CI [1.31–27.95]; $P = 0.02$) were independently and significantly associated with sICH (Table 3). No significant association was observed between lipid subtypes and sICH in the final multivariable regression model. According to the results of the functional outcome analysis, at 3 months, all the patients with sICH ($n = 14$) had unfavorable functional outcomes (mRS ≥ 3). Overall, 103 patients (45%)

**Table 2  Univariate and multivariable regression analysis regarding the occurrence of sICH.**

| | Univariate regression analysis | | | Multivariable regression analysis | | |
|---|---|---|---|---|---|---|
| | OR | 95% CI | *P* value | OR | 95% CI | *P* value |
| Age, per year increase | 1.03 | 0.98–1.08 | 0.21 | | | |
| Male | 1.04 | 0.35–3.09 | 0.95 | | | |
| NIHSS, per point increase | 1.08 | 1.00–1.16 | 0.05 | 1.12 | 1.02–1.22 | 0.01 |
| Time until treatment | 1.001 | 0.99–1.01 | 0.78 | | | |
| Systolic blood pressure | 1.01 | 0.99–1.03 | 0.29 | | | |
| Diastolic blood pressure | 0.98 | 0.93–1.04 | 0.46 | | | |
| Admission serum glucose level | 1.01 | 1.00–1.01 | 0.03 | 1.01 | 0.996–1.02 | 0.26 |
| Total cholesterol level | 0.995 | 0.98–1.01 | 0.46 | | | |
| HDL-C level | 0.997 | 0.95–1.04 | 0.90 | 1.003 | 0.96–1.05 | 0.90 |
| LDL-C level | 0.98 | 0.97–1.00 | 0.10 | 0.99 | 0.97–1.00 | 0.10 |
| Triglyceride level | 1.002 | 0.995–1.01 | 0.63 | 1.00 | 0.99–1.01 | 0.95 |
| Prothrombin time-INR level | 0.68 | 0.05–9.06 | 0.77 | | | |
| Creatinine | 1.45 | 0.4–5.22 | 0.58 | | | |
| *Hemoglobin* A1c | 1.42 | 0.95–2.13 | 0.09 | | | |
| History of arterial hypertension | 1.74 | 0.38–8.02 | 0.48 | | | |
| Diabetes mellitus | 7.00 | 2.11–23.25 | 0.001 | 6.04 | 1.31–27.95 | 0.02 |
| Hyperlipidemia | 0.67 | 0.20–2.21 | 0.51 | | | |
| Atrial fibrillation | 1.43 | 0.49–4.23 | 0.52 | | | |
| Coronary heart disease | 0.98 | 0.70–1.36 | 0.88 | | | |
| Current smoker | 0.30 | 0.04–2.36 | 0.25 | | | |
| Previous stroke | 0.60 | 0.8–4.79 | 0.63 | | | |

Notes.

*HDL-C*, high-density lipoprotein cholesterol; *LDL-C*, low-density lipoprotein cholesterol; *NIHSS*, National Institutes of Health Stroke Scale; *sICH*, symptomatic intracerebral hemorrhage.

**Table 3  Multivariable regression analysis: association of lipid levels with favorable outcomes.**

| | Odds ratio | 95% CI | *P* value |
|---|---|---|---|
| Age | 0.99 | 0.96–1.02 | 0.40 |
| NIHSS | 0.86 | 0.81–0.91 | <0.001 |
| Admission serum glucose level | 0.997 | 0.99–1.00 | 0.38 |
| History of diabetes mellitus | 0.53 | 0.96–1.02 | 0.14 |
| LDL-C level | 1.002 | 0.99–1.01 | 0.67 |
| HDL-C level | 0.99 | 0.96–1.02 | 0.41 |
| Triglyceride level | 1.002 | 0.997–1.01 | 0.46 |

Notes.

*HDL-C*, high-density lipoprotein cholesterol; *LDL-C*, low-density lipoprotein cholesterol; *NIHSS*, National Institutes of Health Stroke Scale.

had favorable outcomes (mRS ≤ 2) at 3 months (Table 1). Lipid subtype levels did not significantly differ between the patients with favorable outcomes and those with unfavorable outcomes. After adjustment for covariates in the multivariate analysis, a higher NIHSS score was found to be independently associated with unfavorable outcomes (OR = 0.86; CI [0.81–0.91]; $P < 0.001$).

## DISCUSSION

The present study investigated the effects of lipid subtype levels on clinical outcomes in patients with ischemic stroke following the intravenous administration of rt-PA. The major finding of this study was that the levels of lipid subtypes, namely total cholesterol, LDL-C, HDL-C, and triglycerides, were not associated with the occurrence of sICH or unfavorable outcomes following rt-PA treatment. The NIHSS score and a history of diabetes mellitus were independently and significantly associated with the risk of sICH. A higher NIHSS score increased the odds for unfavorable outcomes at 3 months.

Our findings are in contrast to those of a previous study that reported an association between low LDL-C levels and sICH (*Bang et al., 2007*). This discrepancy in findings may partly be explained by differences in the study design. All our patients had received intravenous rt-PA. However, Bang et al. investigated 104 patients who received thrombolysis with intravenous or intra-arterial rt-PA and mechanical recanalization. In their study, only three (12%) patients developed sICH following the intravenous administration of rt-PA and the occurrence of sICH was higher in patients (18%) who received intra-arterial rt-PA and mechanical recanalization (18%). Another two studies have identified triglycerides as an independent predictor of sICH in patients with acute ischemic stroke (*Messe et al., 2013*; *Uyttenboogaart et al., 2008*). *Uyttenboogaart et al. (2008)* analyzed 252 patients, all of whom received intravenous rt-PA, and found that a higher NIHSS score, prior antiplatelet use, and a higher triglyceride level were independently associated with sICH. By contrast, *Messe et al. (2013)* included 22,216 patients who received intravenous rt-PA and demonstrated that higher HDL-C and lower triglyceride levels were associated with a moderately increased risk of sICH. The inconsistent association between triglycerides and sICH might be due to the measurement of lipids in different states in these studies. In the study of *Uyttenboogaart et al. (2008)* serum lipid levels were tested in an acute phase and nonfasting state. However, information regarding the time when lipid measurements were performed was not provided in the study conducted by *Messe et al. (2013)*. In our study, we measured fasting lipid levels after the occurrence of acute ischemic stroke. The reason for this discrepancy in triglyceride levels is not clear; however, acute psychological stress was reported to delay serum triglyceride clearance in healthy, middle-aged men and women (*Stoney et al., 2002*).

Our results are compatible with those of the majority of studies that reported that the serum total cholesterol or LDL-C level did not cause an increase in the risk of sICH or poor functional outcomes in acute ischemic stroke following the intravenous administration of rt-PA (*Makihara et al., 2012*; *Nardi et al., 2012*; *Rocco et al., 2012*). Consistent with our results, *Makihara et al. (2012)* investigated 489 patients with acute stroke who received intravenous rt-PA and reported that no relationship existed between any lipid subtype level obtained in the acute phase and sICH, whereas a higher HDL-C level was associated with favorable functional outcomes at 3 months. In addition, *Nardi et al. (2012)* included a large cohort of 1847 patients with acute stroke; they found no association between lipid levels and sICH; however, they reported that low HDL-C and triglyceride levels were independently associated with mortality. *Rocco et al. (2012)* investigated 1,066 patients and reported no

association between the fasting lipid level in the acute phase and sICH in patients treated with rt-PA. The same result was found for mortality or functional outcomes at 3 months. Likewise, our data showed that lipid levels were not related to unfavorable outcomes. A rapid decrease in serum lipid levels is observed in acute illness and is considered to be connected with active inflammation (*Esteve, Ricart & Fernandez-Real, 2005*). Most prior studies that evaluated the association between thrombolysis and sICH have reported results based on serum lipid levels measured at a single time point. However, cholesterol and triglyceride levels are not reliable when not measured in a fasting state and may decrease in the acute phase of stroke (*Phuah et al., 2016*). Furthermore, a temporal association of a decline in serum lipid levels, which was a proxy for increasing systemic inflammation, within 6 months preceding the occurrence of ICH was found in a previous study (*Phuah et al., 2016*). Thus, the discrepancy in the results of prior studies may partly be due to the timing of serum lipid level measurements.

The mechanism underlying the relationship between lipid levels and ICH remains unclear. A meta-analysis that included approximately 800 patients with acute ischemic stroke, most of whom were not treated with intravenous rt-PA, showed that ICH tended to occur more often in patients with lower LDL-C levels, whereas HDL-C and triglyceride levels were not associated with the occurrence of ICH (*Nardi et al., 2011*). By contrast, a recent genetic study found that high serum total cholesterol and LDL-C levels increased the risk of ICH, thereby providing a more reliable lifetime exposure risk (*Akoudad et al., 2016*). In our study, lipid levels were measured only once in the fasting state in the acute phase of stroke. Therefore, we do not have an accurate explanation for how temporal lipid levels affect the occurrence of sICH following thrombolysis. We speculate that serum lipid levels measured in the acute phase of stroke is not an independent risk factor for sICH after thrombolysis in the acute phase of stroke. The results of our study, compatible with the two major risk factors in HAT score, showed that a history of diabetes mellitus and a higher NIHSS score were independently associated with the incidence of sICH after thrombolysis (*Lou et al., 2008*; *Messe et al., 2013*). Furthermore, a history of diabetes mellitus was not associated with unfavorable outcomes at 3 months. This result supports the finding that a history of diabetes mellitus is not a contraindication to intravenous adminsitration of rt-PA (*Mishra et al., 2011*; *Reiter et al., 2014*).

This study has several limitations. This retrospective, observational, case–control study included only post-thrombolytic patients from three hospitals located in Northern Taiwan; this may have introduced selection bias and diminished the generalizability of our results. The sample size of patients who had sICH and poor functional outcomes was small, which limits the use of regression models for the outcome analysis. Moreover, other potential risk factors for sICH, such as early signs of ischemia on brain imaging, recanalization status, and previous statin use, were not consistently evaluated for the entire cohort. However, many studies have reported that previous statin use was not independently associated with sICH following rt-PA treatment (*Makihara et al., 2012*; *Messe et al., 2013*; *Nardi et al., 2012*; *Rocco et al., 2012*; *Uyttenboogaart et al., 2008*). Because of the fact that serum lipid levels were measured only once, the association may be explained by remaining confounding factors due to the presence of undetected factors such as diet

(*Durrington, 1990*). Additionally, information on the body mass index, which was shown to inversely increase the risk of primary ICH, was missing in our study (*Lioutas et al., 2017*). Finally, unmeasured confounding socioeconomic variables, such as the educational status, may have influenced the risk of stroke (*Jackson, Sudlow & Mishra, 2018*).

In summary, total cholesterol, LDL-C, HDL-C and triglyceride levels were not associated with sICH or unfavorable functional outcomes after thrombolysis therapy in acute ischemic stroke. The clinical application requires further research to confirm these results.

## ACKNOWLEDGEMENTS

This manuscript was edited by Wallace Academic Editing.

### Funding

This study was supported by the Taipei Medical University Hospital (104TMUH-NE-01). The funders had no role in study design, data collection and analysis, decision to publish, or preparation of the manuscript.

### Grant Disclosures

The following grant information was disclosed by the authors:
Taipei Medical University Hospital: 104TMUH-NE-01.

### Competing Interests

The authors declare there are no competing interests.

### Author Contributions

- Ting-Chun Lin conceived and designed the experiments, performed the experiments, analyzed the data, contributed reagents/materials/analysis tools, prepared figures and/or tables, authored or reviewed drafts of the paper, approved the final draft.
- Yen-Kuang Lin conceived and designed the experiments, analyzed the data, contributed reagents/materials/analysis tools, prepared figures and/or tables, authored or reviewed drafts of the paper, approved the final draft.
- Chin-I Chen, Lung Chan, Rey-Yue Yuan, Jau-Jiuan Sheu, Chun-Ren Wei and Jui-Ping Tsai contributed reagents/materials/analysis tools.
- Nai-Fang Chi conceived and designed the experiments, analyzed the data, contributed reagents/materials/analysis tools, authored or reviewed drafts of the paper.
- Tu-Hsueh Yeh contributed reagents/materials/analysis tools, authored or reviewed drafts of the paper.

### Ethics

The following information was supplied relating to ethical approvals (i.e., approving body and any reference numbers):

Taipei Medical University-Joint Institutional Review Board (N201512020).

## Data Availability

Raw data is provided in Supplemental Files.

## Supplemental Information

Supplemental information for this article can be found online at http://dx.doi.org/10.7717/peerj.6021#supplemental-information.

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
