# Peer review of "Serum lipid level is not associated with symptomatic intracerebral hemorrhage after intravenous thrombolysis for acute ischemic stroke"

_PeerJ, doi:10.7717/peerj.6021_

## Round 0.1 · original submission · Major Revisions

There are important comments from Peer Reviewer 1 and 3 on defination methodology and analysis.These constitutes major revisions to the manuscript.

Reviewer 1 ·

Basic reporting

This manuscript is overall well organized and professionally written. It can be clearly understood. The statistical data in tables are well described. However, the introduction part could include more related information. There are some issues regarding formatting, but which are minor. Here are some concerns:
1. Even though this study focused on serum lipids, the introduction should include broader background of associated factors to acute ischemic stroke, its treatment options, and how sICH associated with each treatment. When other studies are cited, it is better to specify the sample size of each study, as well as the conclusion. The general words like “some…and others…” should be avoided.
2. Font size in Line 93.
3. “analyses” in Line 121 should be “analysis”.
4. The number of reference in text should be before the period, so the readers know which sentences are cited from references.

Experimental design

The experiments were designed properly to serve the aim of investigating the association between sICH and functional outcome in patients with acute ischemic stroke treated with intravenous. The only issue I found with the description of methods is lacking of information on when the fasting blood was drawn, and the non-fasting blood drawn as well.

Validity of the findings

The study focuses on association between serum lipid profile and sICH after intravenous thrombolysis for acute ischemic stroke. The conclusion is no association. It is a negative result, but it also provides hard evidence to a controversial debate. Here are some thoughts on the discussion of results:
1. How did serum lipids tended to an association for lower sICH rate which authors claimed in Line 133-135? How do they compared to other factors to make the conclusion of tendency?
2. Table 1: To my understanding, the serum lipids were measured in fasting state, while the serum glucose was measured in non-fasting state. Why was serum glucose not recorded in fasting? Please clearly clarify each parameter in fasting or non-fasting state in Table 1.

Additional comments

Overall, it is a study well organized and thoroughly discussed. It provides negative but yet informative results for clinical practice. However, to improve the reading experience of broader readers, the introduction can be expanded to provide more information on the background. There are also some experimental details needs to be specified.

·

Basic reporting

Some grammar mistake should be corrected:
In line 152: “independenlty” should be “independently”.
In line 93, the font size is not consistent.

Experimental design

The experiments are rational designed and the conclusions are well supported by the data.

Validity of the findings

The authors showed that that history of diabetes mellitus is significantly associated with sICH. Moreover, based on univariate regression analysis regarding the occurrence of sICH, the serum glucose level, a critical factor of diabetes, was also significant associated with sICH. This co-relationship was also found by other reports (Mishra, Ahmed et al. 2011, Messe, Pervez et al. 2013, Reiter, Teuschl et al. 2014), however, the authors failed to mention this in the introduction or discuss it in the discussion, which make readers who are not experts in this area hard to comprehend the whole paper. It’s better to add some sentences about the relationship of diabetes and sICH in the discussion.

Additional comments

This finding should be interested by the reader of Peer J while some minor modifications are required before publishing.

Reviewer 3 ·

Basic reporting

The writing of this manuscript is subject to multiple errors in grammar. Seeking help from a professional English writer is strongly recommended.

Experimental design

The definition of "serum lipid level" and "serum lipid profile" are poorly defined---are they looking at associations of individual serum lipids with sICH, or are they looking at it as a whole? From their results, it seems that they are looking at the individuals, while I am not sure whether the results tell anything. I would suggest authors creating an overall score for the lipids level and looking at the real "lipid profile" and sICH, which could be more helpful and the analysis could make more sense.

The authors may need to have more explanations of the variables, including the exposure variables and potential confounding factors included in the analysis.

The authors did not state the current gap on knowledge in the area, which may not prove its meaningfulness.

Validity of the findings

I would not say that the data is well controlled, since there are some important factors not included---such as education, body mass index, drug (e.g., statin) use. If not assessed, it may be included in the limitation.

Also, again, I am not totally convinced by the results of the regression models. Are there potential over-adjustment (e.g., SBP and DBP, different lipids)? How about collinearity?

Additional comments

The manuscript needs some changes before getting published.

---

## Round 0.2 · accepted · Accept

Dear Authors,I am happy to inform that your revised manuscript has been accepted by all 3 peer reviewers.Congratulations as this manuscript will move on for pre-publication processing by Peer J staff.Thanking You.

Reviewer 1 ·

Basic reporting

The revised manuscript has added more current study on acute ischemic stroke and its treatments, which helps to understand the purpose of research carried out in this study. It is well organized and is easy to understand. Literatures were properly cited. Overall, it is an interesting and informative study.

Experimental design

The experiments were designed properly to serve the aim of investigating the association between sICH and functional outcome in patients with acute ischemic stroke treated with intravenous.

Validity of the findings

The study provided a negative result, but it also provided hard evidence to a controversial debate. The authors has changes some wordings to avoid over-stating, which is praiseworthy.

Additional comments

Overall, it is a study well organized and thoroughly discussed. It provides a negative but yet informative results for clinical practice.

·

Basic reporting

The writing of this manuscript is professional.

Experimental design

The author collected the information of 229 patients and methodically analyzed.

Validity of the findings

The finding that "Serum lipid profile is not associated with symptomatic intracerebral hemorrhage after intravenous thrombolysis for acute ischemic stroke" will interesting to both biological and clinical readers.

Additional comments

All my concerns were properly addressed. I think it suits for publishing now.

Reviewer 3 ·

Basic reporting

No comment

Experimental design

No comment

Validity of the findings

No comment

Additional comments

The authors have well addressed my previous comments, and it has been largely improved. I think it looks good now.